# Weight gain after renal transplant: Incidence, risk factors, and outcomes

**Abdulrahman Altheaby** [1,2,3]*, **Nuha Alajlan**[4], **Mohammed F. Shaheen**[1,2,3], **Ghaleb Abosamah**[1,2,3], **Basma Ghallab**[1], **Basayl Aldawsari**[1], **Awatif Rashidi**[1], **Mohammed Gafar**[1], **Ziad Arabi**[1,2,3]

**1** Organ Transplant Center and Hepatobiliary Sciences Department, King Saud bin Abdulaziz University for Health Sciences, King Abdulaziz Medical City, Riyadh, Saudi Arabia, **2** College of Medicine, King Saud bin Abdulaziz University for Health Sciences, Riyadh, Saudi Arabia, **3** King Abdullah International Medical Research Center, Riyadh, Saudi Arabia, **4** Department of Medicine, King Abdulaziz Medical City, King Saud bin Abdulaziz University for Health Science, Riyadh, Saudi Arabia

* a83sa@hotmail.com

## Abstract

### Background

Renal transplantation is the definitive treatment for patients with end-stage renal disease (ESRD). It is associated with better quality of life and patient survival. Nevertheless, these benefits come with rising concerns about weight gain and metabolic abnormalities, which adversely impact transplant outcomes.

### Objective

The objective of this study is to estimate the incidence of weight gain in the first year post-renal transplant in addition to the assessment of potential risk factors and the resulting outcome of the graft.

### Methods

We conducted a single-center retrospective cohort study of all 295 patients who underwent kidney transplantation at King Abdulaziz Medical City (KAMC) between January 2016 and December 2019. Clinical and laboratory variables were collected from electronic records. Continuous variables were reported as mean ± standard deviation. Comparison between groups was assessed by unpaired t-test or Mann-Whitney U test while follow-up data were compared using paired t-test and repeated measures ANOVA. Association between the potential risk factors and the weight gain was assessed by means of binary logistic regression analysis.

### Results

Significant weight gain was observed in 161 (54.6%) patients. Females were 119 (40.30%) of the cohort. The mean age was 45.3±15.1 years. The prevalence of diabetes was 234 (79.6%), while hypertensives constituted 77 (26.3%). The comparison between patients who gained weight significantly and patients with stable weight showed a numerical higher prevalence of female gender in patients who had more weight gain (44.1% vs. 35.8%),

**Data Availability Statement:** All data generated or analyzed during this study are included in the paper.

**Funding:** The authors received no specific funding for this work.

**Competing interests:** The authors have declared that no competing interests exist.

**Abbreviations:** BMI, Body mass index; CMV, Cytomegalovirus; BKV, BK virus; DGF, Delayed graft function; eGFR, Estimated glomerular filtration rate; DM, Diabetes Mellitus; HTN, Hypertension; CAD, Coronary artery disease; FBS, Fasting blood sugar; LDL, Low-density lipoprotein; UTI, Urinary tract infection; ACS, Acute coronary syndrome.

higher diabetes, higher rate of a living donor, and statistically significant lower dialysis duration before transplant. Other clinical and laboratory variables were comparable between the two groups.

## Conclusion

Our study showed a high incidence of clinically significant weight gain among patients post-renal transplantation. Patients with lower dialysis duration, a living kidney donor and those who are obese at baseline were at higher risk of gaining weight. Patients who underwent kidney transplantation should be monitored closely for weight gain and further studies are needed to determine the risk factors and appropriate interventions.

## Introduction

Renal transplantation is the definitive treatment for patients with end-stage renal disease (ESRD). It is associated with better quality of life and patient survival when compared to other renal replacement modalities, such as hemodialysis and peritoneal dialysis [1–3]. Nevertheless, these benefits come with rising concerns about the increased risk of cardiovascular disease in renal transplant recipients due to the development or worsening of hypertension, dyslipidemia, post-transplant new-onset diabetes mellitus, and obesity [4–6].

Obesity is a frequent post-renal transplant complication. Approximately 50% of patients gain weight after renal transplantation, more prominently within the first year, regardless of their pre-transplant nutritional status [7, 8]. In the USA, the prevalence of obesity among kidney transplant recipients had increased to over 30% whereas almost 10% of them had morbid obesity in 2011 as compared to less than 20% in the 1990s [9, 10]. In one study, weight gain in the first year post renal transplantation varied between 6 and 10 kg, reciprocating a mean change in BMI of 2 and 3.8 kg/m$^2$ respectively [11]. Another study, that reviewed the characterization of body composition and fat mass distribution one year after renal transplantation, revealed an increased body mass index, total body fat, and visceral fat [12].

Various studies have been done to investigate the risk factors that may contribute to an increase in weight post-renal transplantation. Female gender, lower pre-transplant weight, younger age, living donor transplant, use of corticosteroids, and poor physical activity have been identified as potential risk factors [13–17]. By the same token, increasing body weight post kidney transplant is associated with undesirable metabolic consequences such as post-transplant diabetes mellitus, and may lead to a higher graft-failure rate and higher mortality [18–20]. An analysis of the United States Renal Data System database showed a U-shaped relationship between patient/graft survival and both extreme weight loss or weight gain post kidney transplantation [21].

Due to the lack of studies evaluating the incidence, risk factors, and outcome of increased BMI post-renal transplantation in Saudi Arabia and the Gulf population, we aimed to study the incidence of weight gain in the first-year post-renal transplant in addition to the assessment of potential risk factors and the resulting outcome of the graft.

## Patients and methods

We conducted a single-center retrospective cohort study of all 295 patients who underwent kidney transplantation at King Abdulaziz Medical City (KAMC), Riyadh, Saudi Arabia between January 2016 and December 2019. We exclude all patients who missed follow up

during this period and patients who experienced a loss of graft function in the first year. BMI was calculated as weight in kilograms divided by the square of the height in meters ($kg/m^2$). The study was reviewed and approved by the Institutional Ethics Review Board of King Abdullah International Medical Research Center (KAIMRC) with Memo Ref. No. IRB/1146/20.

Data were collected from the electronic recording system (BESTcare) used in KAMC. All recipients had a measurement of body weight and height during routine follow up at 1 month, 6 months, and one-year post-transplant in addition to their last BMI before renal transplant. Baseline characteristics including age, gender, comorbidities, past medical history, immuno-suppressant medications, graft function and routine laboratory-based results were documented as well. Type of graft and whether the recipient had rejection or not were also collected from the electronic medical chart.

Patients were divided into two groups; patients who gained more weight defined by an increase BMI $> = 2$ $kg/m^2$ in the first-year post-transplant, and patients with stable weight who did not meet the definition for the first group.

Data were analyzed using IBM SPSS statistics software (version 24.0) (SPSS Inc., Chicago, IL, USA). Continuous variables were reported as mean ± standard deviation. Comparison between groups was assessed by unpaired t-test or Mann-Whitney test while follow-up data were compared using paired t-test. We performed general linear model repeated measures ANOVA to check for within-group effects of the longitudinal change in creatinine, eGFR, BMI, FBS, HbA1c, systolic and diastolic blood pressure throughout study time points. Categorical variables were presented as numbers and percentages and analyzed using the Chi-square or Fisher's exact tests as appropriate. All clinically relevant variables were tested to confirm a lack of multicollinearity among them, then we conducted backward elimination multi-variable logistic regression to assess the potential risk factors and the weight gain after controlling for age and gender. All reported P values are two-sided, and P value less than 0.05 was considered statistically significant.

## Result

Out of the 311 patients who underwent kidney transplantation between January 2016 and December 2019, 295 patients were evaluated and 16 patients were excluded due to age less than <18 years old (n = 9), or lost follow up post-transplant in our transplant center (n = 7). Weight gain was observed in 161 (54.6%) patients and weight remained stable in 134 (45.4%) patients during one year of follow up. The baseline characteristics of the patients are shown in (Table 1). Females were 119, constituting 40.30% of the cohort. The mean age was 45.3±15.1 years. The prevalence of diabetics was 234 (79.6%), hypertensive people constituted 77 (26.3%), and smokers were 24 (8.1%). Only 12 (4.1%) patients had undergone gastric sleeve surgery before kidney transplantation. The majority of renal transplant operations utilized grafts from living donors constituting 227 (76.9%). As for induction therapy, 185 (62.7%) of patients received anti-thymocyte globulin (ATG). All patients had the same immunosuppressive maintenance regimen including prednisolone 5 mg daily, tacrolimus and mycophenolate mofetil. The assessment of the functional status of our cohort using the Eastern Cooperative Oncology Group (ECOG) performance status score showed that 86.30% were fully active and able to carry on all pre-disease performance without restriction (ECOG 0) (Table 1).

The comparison between patients who gained weight significantly (i.e., BMI increased more than 2 $kg/m^2$) and patients with stable weight showed that most variables were comparable except for the numerical higher prevalence of female gender in patients who had more than 2 $kg/m^2$ BMI increment (44.1% vs. 35.8%), higher living donor, higher diabetes mellitus, and statistically significant lower dialysis duration before transplantation.

**Table 1. Baseline characteristics.**

| Characteristic | Total (n = 295) n (%) | Gained Weight (n = 161) n (%) | Stable Weight (n = 134) n (%) | P value |
|---|---|---|---|---|
| Age [a] | 45.3±15.1 | 45.0±14.6 | 45.7±15.7 | 0.751 |
| Gender [b] | | | | |
| Female | 119 (40.30) | 71 (44.10) | 48 (35.80) | 0.155 |
| Male | 176 (59.70) | 90 (55.90) | 86 (64.20) | |
| Non-Smoker [b] | 258 (87.50) | 139 (86.30) | 119 (88.80) | |
| Ex-smoker | 13 (4.40) | 10 (6.20) | 3 (2.20) | 0.262 |
| Smoker | 24 (8.10) | 12 (7.50) | 12 (9.00) | |
| DM [b] | 77 (26.30) | 44 (27.50) | 33 (24.80) | 0.689 |
| HTN [b] | 234 (79.60) | 129 (80.10) | 105 (78.90) | 0.885 |
| CAD [b] | 39 (13.40) | 19 (11.90) | 20 (15.20) | 0.490 |
| Gastric sleeve surgery [b] | 12 (4.10) | 8 (5.00) | 4 (3.00) | 0.557 |
| Duration of dialysis in years [a] | 2.3±2.4 | 2.0±2.2 | 2.6±2.6 | 0.032 |
| ECOG [b] | | | | |
| 0 | 251 (86.30) | 143 (89.90) | 108 (81.80) | |
| 1 | 32 (11.00) | 14 (8.80) | 18 (13.60) | 0.083 |
| 2 | 8 (2.70) | 2 (1.30) | 6 (4.50) | |
| Donor [b] | | | | |
| Living | 227 (76.90) | 131 (81.40) | 96 (71.60) | 0.053 |
| Deceased | 68 (23.10) | 30 (18.60) | 38 (28.40) | |
| Induction [b] | | | | |
| ATG | 185 (62.70) | 98 (60.90) | 87 (64.90) | 0.546 |
| Basiliximab | 110 (37.30) | 63 (39.10) | 47 (35.10) | |

[a] The P value was calculated by the unpaired t-test.

[b] The P value was calculated by the Fisher's exact test.

The longitudinal follow-up data of BMI and laboratory variables for one year are presented in (Table 2). The bassline BMI for the first group was lower than the second group, (25.5±5.1 vs. 26.7±5.2, respectively; p = 0.051). The weight gain difference started to become evident from the second half of the first year onward.

Creatinine, and similarly eGFR, was better in the first group after one week and significantly better after 1 and 6 months. When tested for the change across the four time points of the study using general linear model repeated measures ANOVA, no significant within group effects were detected in both groups, and no interaction between group and time as well. Other parameters such as FBS, HbA1c and LDL were comparable. Regarding blood pressure, analysis with repeated measures ANOVA for the readings at one, six, and twelve months showed no significant within group effects in both groups for systolic and diastolic BP over the study period (Figs 1 and 2).

The list of complications that occurred in both groups is shown in (Table 3). There was no significant difference in the incidence of biopsy-proven allograft rejection or delayed graft function (DGF) which was defined as the requirement of dialysis in the first week post-transplant. Urological complications such as urine leak and wound infection were comparable as well, nevertheless, cytomegalovirus (CMV) viremia, as detected by screening PCR, was significantly lower in patients with higher weight gain, p = 0.007. Moreover, no significant differences were detected in new-onset diabetes after transplant (NODAT) and mortality rates.

We conducted multivariate backward selection logistic regression analysis to identify potential risk factors associated with weight gaining. The best-fit model is presented in

**Table 2. Metabolic & laboratory variables.**

| Characteristic | Total | Gained Weight | Stable Weight | P value |
|---|---|---|---|---|
| BMI [a] | | | | |
| Pre | 26.0±5.2 | 25.5±5.1 | 26.7±5.2 | 0.051 |
| Post 1 month | 26.0±5.2 | 26.0±5.3 | 26.0±5.2 | 0.989 |
| Post 6 months | 27.5±5.3 | 28.4±5.2 | 26.3±5.1 | 0.001 |
| Post 12 months | 28.6±5.6 | 30.7±5.5 | 26.7±5.2 | <0.001 |
| Creatinine umol/L [a] | | | | |
| 1 week | 111.5±65.1 | 105.3±55.3 | 118.8±74.7 | 0.077 |
| 1 month | 102.3±36.8 | 98.0±34.2 | 107.3±39.2 | 0.031 |
| 6 months | 97.9±33.1 | 92.8±29.0 | 103.9±36.6 | 0.004 |
| 12 months | 93.5±29.7 | 90.7±26.5 | 96.9±33.0 | 0.073 |
| eGFR ml/min [a] | | | | |
| 1 week | 73.6±27.9 | 76.1±28.6 | 70.6±26.9 | 0.089 |
| 1 month | 73.9±22.5 | 76.5±22.8 | 70.8±21.73 | 0.031 |
| 6 months | 75.8±21.1 | 78.4±20.7 | 72.7±21.2 | 0.020 |
| 12 months | 79.5±20.6 | 80.4±20.2 | 78.4±21.1 | 0.402 |
| Proteinuria [*] [b] N (%) | | | | |
| 1 month | 88 (29.80) | 46 (28.60) | 42 (31.30) | 0.612 |
| 6 months | 80 (27.10) | 36 (22.40) | 44 (32.80) | 0.049 |
| 12 months | 72 (24.40) | 37 (23.00) | 35 (26.10) | 0.587 |
| HbA1c [b] N (%) | | | | |
| <5.7% | 189 (93.60) | 100 (93.50) | 89 (93.70%) | 1 |
| 5.7%-6.9% | 32 (71.10) | 17 (70.80) | 15 (71.40) | 1 |
| >6.9% | 59 (81.90) | 35 (83.30) | 24 (80.00) | 0.763 |
| HbA1c (%) [a] | | | | |
| 3 months | 6.4±1.6 | 6.4±1.6 | 6.4±1.6 | 0.909 |
| 6 months | 6.7±1.7 | 6.6±1.6 | 6.7±1.8 | 0.656 |
| 12 months | 6.8±1.8 | 6.8±1.8 | 6.8±1.7 | 0.825 |
| FBS (mmol/L) [b] N (%) | | | | |
| <5.5 | 165 (92.20) | 85 (88.50) | 80 (96.40) | 0.091 |
| 5.5–7 | 48 (77.40) | 25 (69.40) | 23 (88.50) | 0.123 |
| >7 | 67 (82.70) | 39 (78.00) | 28 (90.30) | |
| FBS (mmol/L) [a] | | | | |
| 1 month | 7.0±3.6 | 7.3±4.1 | 6.7±2.8 | 0.188 |
| 6 months | 7.0±3.4 | 7.1±3.4 | 6.9±3.4 | 0.498 |
| 12 months | 6.8±3.3 | 7.0±3.6 | 6.6±2.9 | 0.369 |
| LDL (mmol/L) [a] | | | | |
| Pre | 2.6±0.8 | 2.6±0.8 | 2.5±0.8 | 0.294 |
| 3 months | 2.5±0.9 | 2.5±0.8 | 2.6±1.0 | 0.602 |
| 12 months | 2.5±0.9 | 2.5±0.7 | 2.5±1.0 | 0.869 |

[*]Proteinuria = Urine albumin creatinine ratio (ACR) of 3 mg/mmol or more.

[a] The P value was calculated by the unpaired t-test.

[b] The P value was calculated by the Fisher's exact test.

Table 4. Only living donor [OR = 1.80; 95%CI (1.02 to 3.18); p = 0.043] and baseline BMI > = 30 [OR = 0.572; 95%CI (0.32 to 0.99); p = 0.048] were found significant predictors in our model.

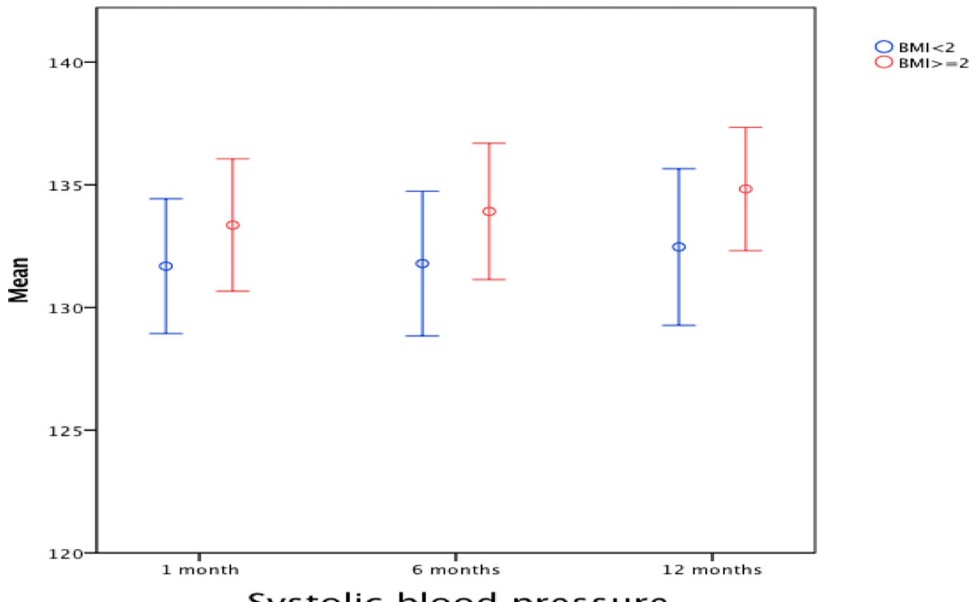

**Fig 1. Mean and 95% confidence interval of the systolic BP (mmHg) in the first year of the study in both groups (BMI < 2 vs. BMI > = 2).**

## Discussion

This retrospective cohort study aimed to assess the incidence, risk factors, and outcomes of weight gain among patients who underwent renal transplantation in King Abdulaziz Medical City (KAMC) from January 2016 to December 2019 during the first year of follow up. The incidence of significant weight gain in the first year after renal transplantation was estimated

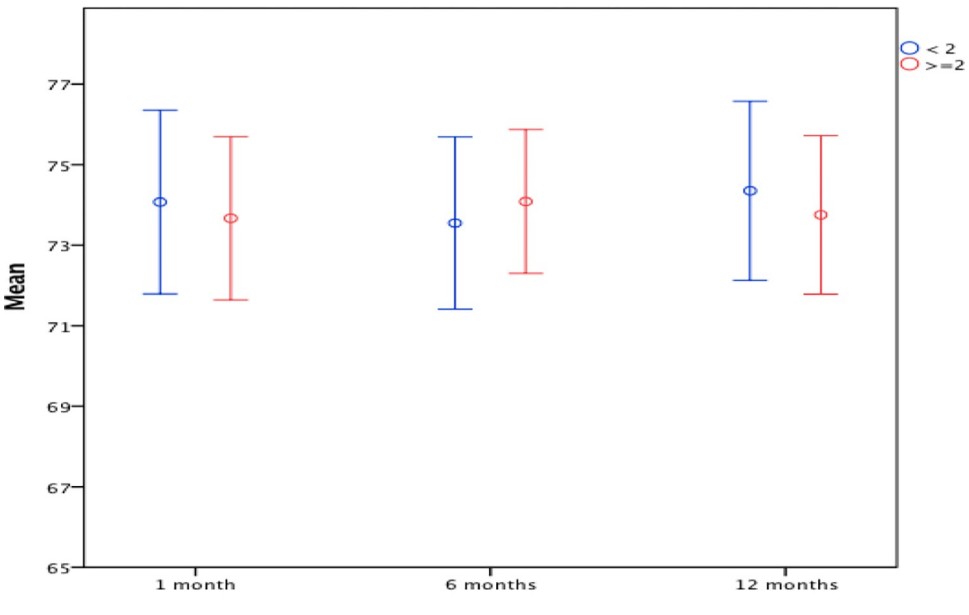

**Fig 2. Mean and 95% confidence interval of the diastolic BP (mmHg) in the first year of the study in both groups (BMI < 2 vs. BMI > = 2).**

**Table 3. Complications.**

| Characteristic | Total n (%) | Gained Weight n (%) | Stable Weight n (%) | P value |
|---|---|---|---|---|
| Rejection [a] | 29 (9.80) | 18 (11.20) | 11 (8.20) | 0.437 |
| DGF [a] | 18 (6.10) | 9 (5.60) | 9 (6.80) | 0.808 |
| Wound infection [a] | 6 (2.00) | 4 (2.50) | 2 (1.50) | 0.693 |
| Urine leak [a] | 3 (1.00) | 2 (1.20) | 1 (0.80) | 1 |
| UTI [a] | 79 (26.80) | 41 (25.50) | 38 (28.40) | 0.599 |
| NODAT [a] | 14 (4.80) | 7 (4.30) | 7 (5.30) | 0.787 |
| CMV viremia [a] | 158 (53.70) | 75 (46.60) | 83 (62.40) | 0.007 |
| BK viremia [a] | 52 (17.70) | 29 (18.00) | 23 (17.30) | 0.880 |
| Arrhythmia [a] | 7 (2.40) | 2 (1.20) | 5 (3.70) | 0.251 |
| ACS [a] | 3 (1.00) | 0 (0.00) | 3 (2.20) | 0.093 |
| Mortality [a] | 14 (4.70) | 6 (3.70) | 8 (6.00) | 0.418 |
| Started insulin [a] | 8 (2.70) | 5 (3.10) | 3 (2.20) | 0.732 |

[a] The P value was calculated by Fisher's exact test.

as (54.6%) in our cohort. This is similar to other studies which showed a high incidence of weight gain post-renal transplantation.

In our study, we found a numerical higher prevalence of female gender (44.1% vs. 35.8%), higher living donor, and higher diabetic patients in kidney transplant recipients who had more than 2 kg/m$^2$ BMI increment, but it was not statistically significant. On the other hand, statistically significant weight gain was only observed in patients with lower dialysis duration before transplant. This can be explained by the poor nutritional status of patients on prolonged dialysis periods as they need longer recovery time post-transplantation. A study in Poland reported that BMI was increased in almost 65% of kidney recipients, and the weight gain was higher in patients with normal BMI pre-transplantation compared to those classified as overweight or obese [22]. Additionally, our findings did not show that females were at higher risk to gain weight, unlike other reported studies that have estimated their risk to be up to two times higher than males [16, 23].

We examined related laboratory parameters at specific intervals from kidney transplant throughout the first year then compared the findings between the group who gained more weight and those with stable weight. Acknowledging the limitation of the small number of patients and length of the follow-up, we did not notice significant differences in those parameters. In contrast, Aminu et al. have reported that obese patients post-renal transplantation were more prone to graft dysfunction and progression of atherosclerosis, including higher mean arterial blood pressure, total cholesterol, triglycerides, left ventricular mass index, and higher carotid intima-media thickness [24]. This could be explained by the fact that the mean duration of follow-up in the Aminu et al. study was 60 ± 18.8 (range 36–89) months while it was 12 months in our study.

**Table 4. Multivariate analysis for risk factors.**

| Predictor | P value | Odds ratio | 95% CI lower | 95% CI upper |
|---|---|---|---|---|
| Age > 40 | 0.714 | 1.104 | 0.651 | 1.872 |
| Female sex | 0.080 | 1.559 | 0.948 | 2.563 |
| Pre Transplant BMI > = 30 | 0.048 | 0.572 | 0.328 | 0.996 |
| DM | 0.432 | 1.266 | 0.703 | 2.281 |
| Living donor | 0.043 | 1.802 | 1.02 | 3.184 |

Our study has not shown any statistically significant differences in biopsy-proven allograft rejection, wound infection, UTI or urine leak between the two groups. Although prior study done in our center have shown that higher baseline BMI was associated with a higher risk of delayed graft function (DGF) [25], our results failed to show deterioration of graft function during the first year of follow up despite weight gain. However, another study showed a statistically significant increase in graft loss in patients with a 5% increase in BMI in the first-year post-transplantation [23]. Interestingly, we found that the frequency of CMV viremia was lower in patients who had more weight gain (p = 0.007), despite the fact that all our recipients were seropositive for CMV IgG, and they have been maintained on valganciclovir 450 mg daily dose for 3 months post-transplantation as per the standard protocol in our center. We could not find an explanation for this finding yet this is consistent with Cristina et al. findings, where patients who gained more than 5% of weight had a lower rate of CMV infection [23].

The data on the association between new-onset diabetes post kidney transplantation (NODAT) and weight gain are controversial. A meta-analysis study, done in 2018 to check the correlation between NODAT and increasing BMI post-transplant, showed that increased BMI is an independent risk of having NODAT [26]. Furthermore, a recent study, done in 2020, showed that post-transplant obesity was associated with a higher rate of type 2 DM and NODAT [27]. Nevertheless, another study showed that higher BMI patients are more likely to develop NODAT, yet patients with NODAT had lower weight gain post-transplant [28].

In contrast, our study has not shown any significant relationship between increasing BMI and developing NODAT (P = 0.787).

In our study, we did not include the effect of exercise, nutritional status and types of various diets. A previous randomized controlled trial implemented an intensive diet versus a standard diet in patients with renal transplantation to examine the impact on weight gain post-transplant. The study demonstrated no significant difference between the two groups. instead, the increase of the BMI in both groups was <5%, which was lower than other studies where the increase was equal to or more than10% [29]. This suggests that dietary and exercise measures are beneficial in avoiding excessive weight gain among kidney transplant patients.

Our study has several limitations; it is a retrospective study with a relatively short follow-up period for the detection of complications that may take years to develop. Additionally, we did not investigate all variables related to weight and metabolic outcomes such as body composition, exercise role, dietary habits, and family history of obesity. However, we have been able to show the high incidence of weight gain post kidney transplantation and identify some risk factors. A prospective study would be able to illustrate the risk factors of weight gain post-transplantation and guide the interventions to prevent it.

## Conclusion

Our study showed a high incidence of weight gain among patients post-renal transplantation in the first year of follow-up. Patients with lower dialysis duration before transplant, receiving living donor graft and those who are obese at baseline were at higher risk of gaining weight. We have not been able to present evidence of the association between graft dysfunction during the period of follow up and weight gain. Overall, post kidney transplantation patients should be monitored closely for weight gain, and further studies are needed to determine the risk factors and appropriate interventions.

## Author Contributions

**Data curation:** Abdulrahman Altheaby, Nuha Alajlan, Basma Ghallab, Basayl Aldawsari, Awatif Rashidi, Mohammed Gafar.

**Formal analysis:** Awatif Rashidi.

**Methodology:** Abdulrahman Altheaby, Mohammed F. Shaheen, Mohammed Gafar.

**Supervision:** Abdulrahman Altheaby, Ziad Arabi.

**Writing – original draft:** Abdulrahman Altheaby, Nuha Alajlan.

**Writing – review & editing:** Abdulrahman Altheaby, Mohammed F. Shaheen, Ghaleb Abosamah, Ziad Arabi.

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
