## [Decision Letter · Decision Letter 0]

8 Nov 2021

PONE-D-21-31746Weight Gain After Renal Transplant: Prevalence, Risk Factors, and OutcomesPLOS ONE

Dear Dr. Altheaby,

Thank you for submitting your manuscript to PLOS ONE. After careful consideration, we feel that it has merit but does not fully meet PLOS ONE’s publication criteria as it currently stands. Therefore, we invite you to submit a revised version of the manuscript that addresses the points raised during the review process.

We look forward to receiving your revised manuscript.

Kind regards,

John Richard Lee, M.D.

Academic Editor

PLOS ONE

Journal Requirements:

Additional Editor Comments;

In addition to the reviewers’ comments, please consider the following:

1) It’s unclear to me that weight gain was observed in 235 patients and weight decreased in 57 patients but Table 1 shows gained weight (n=161) and stable weight (n=134).

2). How did the authors choose a BMI >2 as a cutoff? Do the authors have literature support this cutoff?

3) Table 1 should have legend explaining the test comparing the two groups (i.e. Fisher’s exact test and/or unpaired t-test). Also Table 2 and Table 3.

4) What were the maintenance immunosuppression in each group? This data should be included. Was the weight gain driven by prednisone? Are differences in other regimens?

5) Is the repeated measured analysis with contrasts? Can you please explain more detail in the methods how this was done since you are comparing groups at different time points?

6) How did you define proteinuria? Did you quantify the proteinuria with a Urine P/C ratio?

7) More detail is needed in Table 3: how did you define rejection, DGF, CMV, and BK? How long was the follow up for patients, was this all at 1 year?

8) In the multivariable logistic regression, how did you choose the variables that went into the multivariable analysis? Did you do a univariate analysis and those significant went into the multivariable analysis? How did you come to choose age > 40 rather than continuous variable?

9). The authors note mortalities? Did this occur in the first year? If so, how was the weight account for if the patient died before 1 year?

10) Figure 1 and Figure 2 need legends and should be separated

11) The manuscript would benefit from grammatically editing

Reviewers' comments:

Reviewer's Responses to Questions

**Comments to the Author**

1. Is the manuscript technically sound, and do the data support the conclusions?

Reviewer #1: Yes

Reviewer #2: No

2. Has the statistical analysis been performed appropriately and rigorously? 

Reviewer #1: I Don't Know

Reviewer #2: No

3. Have the authors made all data underlying the findings in their manuscript fully available?

Reviewer #1: Yes

Reviewer #2: Yes

4. Is the manuscript presented in an intelligible fashion and written in standard English?

Reviewer #1: Yes

Reviewer #2: No

5. Review Comments to the Author

Reviewer #1: This is a single center retrospective cohort study of 295 kidney transplant recipients in Saudi Arabia whose objective is to estimate the prevalence of weight gain in the first-year post kidney transplantation, identify potential risk factors, and assess allograft outcomes. The authors observe weight gain in nearly 80% of patients, with patients with lower dialysis duration having significantly higher risk of weight gain. Graft outcomes were comparable in both the “stable weight” and “gained weight” groups. As noted by the authors, a limitation is the short follow up period for potential complications.

Overall, the authors’ findings will be beneficial to the field, adding to data regarding the prevalence and significance of weight gain post renal transplantation, particularly in a population from Saudi Arabia that has not been well studied. The manuscript is organized and written reasonably well. However, it requires careful review and proofreading of grammatical and typographical errors. My recommendations include the following:

1. The authors report that patients “had the same tacrolimus-based maintenance immunosuppression regimen”. It would be useful to clarify if patients are on steroid-free or steroid maintenance immunosuppression and report if there are differences in steroid use between the groups.

2. Abbreviations and acronyms such as CMV, BK, DM, FBS should be spelled out in full the first time they are used, and the standard abbreviation provided in parentheses.

3. In the Results section there is mention of 57 patients having decreased weight. Were these patients categorized under the “stable weight” group?

4. In Table 1, please report units of measurement for “Duration of dialysis” (e.g. months or years).

5. In Figures 1 and 2, please report units of measurement for blood pressure. In Figure 2, please label the two groups appropriately by adding “BMI” to “<2” and “>=2”.

6. In Table 2, please clarify criteria for “proteinuria”.

7. In Table 2, “FBS” is reported as mg/dL, however the values provided appear to be consistent with mmol/L.

8. In Table 3, please define “BK”. For instance, is this referring to BKV nephropathy, BKV viremia or BKV viruria?

9. The authors report CMV infection was significantly less common in the group that gained more weight, and they could not find an explanation for this finding. However, they do not discuss any further analysis that may have been done. It may be useful to explore variables such as CMV serostatus of the donor/recipient pair, CMV prophylaxis regimen, and donor type (there was a trend toward more deceased donors in the "stable weight" group).

10. There are numerous typographical and grammatical errors throughout the manuscript. Recommend a meticulous review and proofreading of the manuscript.

Thank you for the opportunity to review this manuscript.

Reviewer #2: 1. In methods, clarify "Comparison between groups was assessed by unpaired t-test or Mann-Whitney test while

follow-up data were compared using paired t-test and repeated measures ANOVA". what does groups and follow up data refer to? Also, express age as median and interquartile range since it is a skewed distribution.

2. The induction therapy in about 60% was ATG, what was the regimen in the remaining? What is the tacrolimus based regimen- does it include steroids? Need to clarify the immunosuppresive regimen

3. Many descriptions used in the table is ambiguous- what does "bariatric" refer to? What is the unit used to express duration of dialysis. What does DM, HTN and CAD stand for. Similarly table 2 also needed to be revised to correct for incomplete descriptions and units. What was the definition of CMV infection used for the manuscript?

4. Many abbreviations are used throughout the manuscript without mentioning their expansion in the manuscript.

5. Since the authors have not rigorously studied if the outcomes were directly related only to the weight gained post transplant and it is a retrospective study, it is not appropriate to use the term "outcomes" in the title and conclusion

6. Since the authors looked at weight gain post transplantation, it is looking at incidence rather than prevalence of weight gain

7. The manuscript needs to rigorously reviewed again for several grammatical errors and poor sentence construction

6. PLOS authors have the option to publish the peer review history of their article (what does this mean?). If published, this will include your full peer review and any attached files.

Reviewer #1: No

Reviewer #2: No

---

## [Author Response · Author response to Decision Letter 0]

6 Jan 2022

1. All patients had the same tacrolimus-based maintenance immunosuppression regimen which include Prednisolone 5 mg daily, Tacrolimus and mycophenolate.

2. Abbreviations added to revised manuscript. 

3. corrected in revised manuscript. 

4. Duration of dialysis counted by years. clarified in revised manuscript.

5. units of measurement for blood pressure. done 

6. definition of proteinuria added to revised manuscript.

7. FBS” is reported as mg/dL, corrected with mmol/L.

8. define “BK”= means BK viremia . 

9. CMV infection : CMV serostatus and prophylaxis regimen has been added to revised manuscript. 

detail answer added to respond to reviewer.

---

## [Decision Letter · Decision Letter 1]

17 Feb 2022

PONE-D-21-31746R1Weight Gain After Renal Transplant: incidence, Risk Factors, and OutcomesPLOS ONE

Dear Dr. Altheaby,

Thank you for submitting your manuscript to PLOS ONE. After careful consideration, we feel that it has merit but does not fully meet PLOS ONE’s publication criteria as it currently stands. Therefore, we invite you to submit a revised version of the manuscript that addresses the points raised during the review process.

We look forward to receiving your revised manuscript.

Kind regards,

John Richard Lee, M.D.

Academic Editor

PLOS ONE

Journal Requirements:

Additional Editor Comments:

Please address comments of the reviewer.

1) Also please put the details in the methods/results on what was done specifically for the anova testing, as described in the response letter.

2) Also please put the details in the methods/results on what was done specifically for the multivariable logistic regression, as described in the response letter.

3) The manuscript needs thorough editing for fluency and grammar as also indicated by Reviewer 1.

Reviewers' comments:

Reviewer's Responses to Questions

**Comments to the Author**

1. If the authors have adequately addressed your comments raised in a previous round of review and you feel that this manuscript is now acceptable for publication, you may indicate that here to bypass the “Comments to the Author” section, enter your conflict of interest statement in the “Confidential to Editor” section, and submit your "Accept" recommendation.

Reviewer #1: (No Response)

Reviewer #2: All comments have been addressed

2. Is the manuscript technically sound, and do the data support the conclusions?

Reviewer #1: Yes

Reviewer #2: Yes

3. Has the statistical analysis been performed appropriately and rigorously? 

Reviewer #1: I Don't Know

Reviewer #2: Yes

4. Have the authors made all data underlying the findings in their manuscript fully available?

Reviewer #1: Yes

Reviewer #2: Yes

5. Is the manuscript presented in an intelligible fashion and written in standard English?

Reviewer #1: Yes

Reviewer #2: Yes

6. Review Comments to the Author

Reviewer #1: Thank you for the revisions and helpful clarifications.

1. There are several typographical errors throughout the manuscript. Again, recommend a thorough review and editing of grammatical errors. For instance, in the Abstract under “Objective”, the sentence should likely read, “The objective of this study is to estimate…”

2. In the multivariate analysis (Table 4), it may be helpful to state why and how age >40 was chosen as a variable.

3. For Figures 1 and 2, please add units of measurement for blood pressure.

Thank you for the opportunity to review this manuscript.

Reviewer #2: (No Response)

7. PLOS authors have the option to publish the peer review history of their article (what does this mean?). If published, this will include your full peer review and any attached files.

Reviewer #1: No

Reviewer #2: No

---

## [Author Response · Author response to Decision Letter 1]

7 Apr 2022

Hi

thank you for reviewing my manuscript. 

correction has been made .

thank you

---

## [Editor Report · Decision Letter 2]

21 Apr 2022

Weight Gain After Renal Transplant: incidence, Risk Factors, and Outcomes

PONE-D-21-31746R2

Dear Dr. Altheaby,

We’re pleased to inform you that your manuscript has been judged scientifically suitable for publication and will be formally accepted for publication once it meets all outstanding technical requirements.

Kind regards,

John Richard Lee, M.D.

Academic Editor

PLOS ONE
---

## [Editor Report · Acceptance letter]

23 May 2022

PONE-D-21-31746R2 

Weight Gain After Renal Transplant: Incidence, Risk Factors, and Outcomes 

Dear Dr. Altheaby:

I'm pleased to inform you that your manuscript has been deemed suitable for publication in PLOS ONE. Congratulations! Your manuscript is now with our production department. 

Kind regards, 

on behalf of

Dr. John Richard Lee 

Academic Editor

PLOS ONE